# SUPERVISED CHAIN OF THOUGHT

## ABSTRACT

Large Language Models (LLMs) have revolutionized natural language processing and hold immense potential for advancing Artificial Intelligence. However, the core architecture of most mainstream LLMs—the Transformer—has inherent limitations in computational depth, rendering them theoretically incapable of solving many reasoning tasks that demand increasingly deep computations. Chain of Thought (CoT) prompting has emerged as a technique to address these architectural limitations, as evidenced by several theoretical studies. It offers a promising approach to solving complex reasoning tasks that were previously beyond the capabilities of these models. Despite its successes, CoT and its variants (such as Tree of Thought, Graph of Thought, etc.) rely on a "one-prompt-for-all" approach, using a single prompt structure (e.g., "think step by step") for a wide range of tasks—from counting and sorting to solving mathematical and algorithmic problems. This approach poses significant challenges for models to generate the correct reasoning steps, as the model must navigate through a vast prompt template space to find the appropriate template for each task. In this work, we build upon previous theoretical analyses of CoT to demonstrate how the one-prompt-for-all approach can negatively affect the computability of LLMs. We partition the solution process into two spaces: the prompt space and the answer space. Our findings show that task-specific supervision is essential for navigating the prompt space accurately and achieving optimal performance. Through experiments with state-of-the-art LLMs, we reveal a gap in reasoning performance when supervision is applied versus when it is not. Our goal is to provide deeper insights into the mechanisms underlying CoT, offering guidance for the effective design of CoT variants. Additionally, we underscore the limitations of traditional "unsupervised" prompting methods, arguing that users of CoT cannot simply "sit back" and rely entirely on the model. Instead, we advocate for task-specific "supervised" CoT, enriched with human knowledge, to enable more effective reasoning in LLMs.

## 1 INTRODUCTION

The advent of large language models (LLMs) (Achiam et al., 2023) has ushered in a new era for natural language processing and artificial intelligence (Kojima et al., 2022; Zhao et al., 2023). These models exhibit remarkable capabilities across various domains (Thirunavukarasu et al., 2023; Wei et al., 2022; Valmeekam et al., 2023; Zhang et al., 2023), achieving near-human performance in tasks such as knowledge retrieval and articulation (Chang et al., 2024). However, concerns have been raised regarding their reasoning abilities (Valmeekam et al., 2022; Zhang et al., 2024). These tasks range from fundamental operations like counting, sorting, and multiplication (Dziri et al., 2024), to more complex challenges such as mathematical problem-solving, algorithm design, and coding (Xu et al., 2022; Thirunavukarasu et al., 2023). Previous research has explored several factors contributing to these reasoning deficiencies, including training optimizations (Thorburn & Kruger, 2022), tokenization methods (Singh & Strouse, 2024), and dataset choices (Ye et al., 2024). Among these, the architecture of the model plays a pivotal role in determining its reasoning capabilities (Raghu et al., 2017; Zhang et al., 2024; Delétang et al., 2022). The backbone architecture of most mainstream LLMs—the Transformer (with finite precision) (Vaswani, 2017)—has intrinsic limitations related to computational depth (Li et al., 2024). Specifically, the attention mechanism within Transformers can perform only a fixed number of sequential computational steps (Li et al., 2024; Zhang et al., 2024; Sanford et al., 2024; Dehghani et al., 2018), leading to *constant-depth* modeling (Li et al., 2024). As a result, when relying solely on the Transformer's *internal reasoning*, the model's

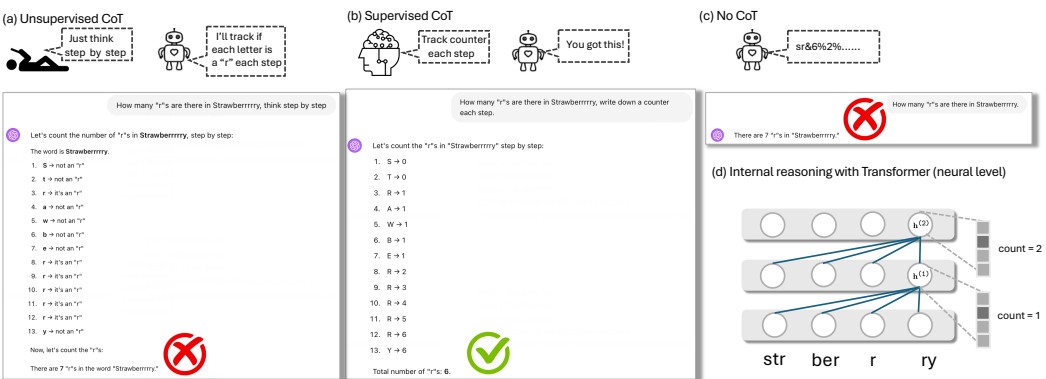

Figure 1: **(a)** Without supervision during CoT, the model generates its own step template for recurrent computation. This template can be incorrect, leading to task failure. **(b)** With human supervision, the task performance under CoT can be properly guided. **(c)** When CoT is not employed, the model relies solely on its internal reasoning via the Transformer architecture. **(d)** The Transformer can only perform constant-depth sequential computations. We assume that this Transformer neither *memorizes* the results nor performs bit-level (circuit) reasoning; instead, reasoning occurs at the neuron (hidden state) level.

computability is restricted to $TC^0$ (Li et al., 2024; Feng et al., 2024), which confines it to solving tasks of limited complexity and length (Figure 1.c-d).

The emergence of Chain of Thought (CoT) (Wei et al., 2022) has transformed the computational limitations imposed by architectural design. Specifically, it overcomes the *constant depth* limitation of a model's internal (latent space) reasoning by extending the reasoning process into the "text" space through prompting (Li et al., 2024; Zhang et al., 2024; Feng et al., 2024). As demonstrated by both theoretical (Li et al., 2024; Zhang et al., 2024; Feng et al., 2024) and empirical research (Li et al., 2024; Zhang et al., 2024), CoT effectively enhances the reasoning depth of Transformer-based models (with finite precision), achieving "Turing Completeness" under ideal conditions (Li et al., 2024; Zhang et al., 2024). While the theoretical analysis of CoT focuses on the model's upper bound computational power, which may not always align with real-world performance, a deep understanding of the CoT mechanism—particularly how it breaks architectural constraints—is crucial for designing better prompts that maximize the computational potential of real-world LLMs. This understanding also forms the basis for our analysis of "supervised" CoT and our prompt search space theories. Therefore, our work first revisits the underlying mechanisms of CoT from a computational perspective, integrating insights from prior research (Li et al., 2024; Zhang et al., 2024; Feng et al., 2024) to demystify them in a clear, yet comprehensive manner, enriched by our unique perspective.

Although theoretical analysis has proven the *existence* of solutions for (almost) any problem using CoT, based on computability and Turing Completeness theory, the actual *discovery* of those solutions can be much more challenging. This is akin to how a Turing machine *can* model solutions for any problem (Boolos et al., 2002) but *finding* the exact Turing machine for a specific NP problem could be difficult. These challenges arise from two main factors for LLMs with CoT. First, the model must develop the correct "step-by-step" template, which essentially embodies the algorithm used for solving the problem (Figure 1.a-b). For instance, the "steps" for solving a graph search problem using depth-first search (DFS) differ from those of a breadth-first search (BFS) algorithm. Second, even after the template (algorithm) is established, finding the solution might require extensive reasoning and exploration to achieve the optimal outcomes. For example, using the BFS template to locate a target node in a tree involves traversing multiple paths in the search space that can be computationally expensive and error-prone.

The vanilla design of CoT is "unsupervised", meaning that the model generates its step template without task-specific supervision from humans. Specifically, when prompted to "think step by step", LLMs autonomously generate a step template (algorithm) it needs to follow—for instance, generating previously visited paths at each step—and then proceeding to search for answers based on this self-generated template (Figure 1.a). Clearly, this naive CoT approach can lead to poor performance,

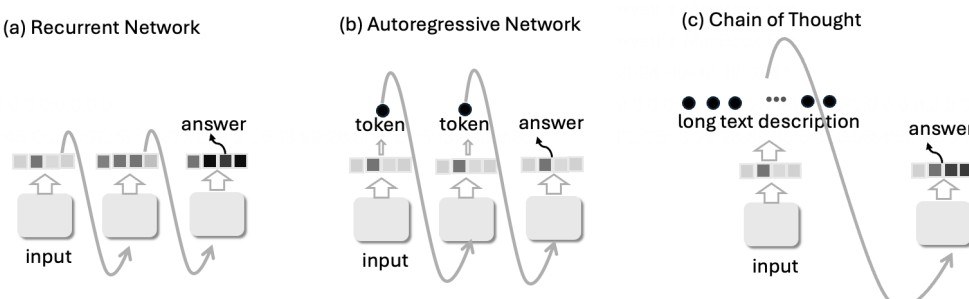

Figure 2: Comparison between recurrence and autoregression.

as the model may generate sub-optimal step templates (algorithms), which hinder the search process. For example, a problem requiring DFS might be unnecessarily attempted with a BFS template generated by the vanilla CoT, incurring high inference costs and likely delivering incorrect answers (Figure 1.a).

Variants of Chain of Thought, such as Tree-of-Thought (Yao et al., 2024) and Graph-of-Thought (Besta et al., 2024), aim to improve the search process within the *answer space*, rather than the prompt space, and remain *unsupervised*. These "X-of-thought" approaches still rely on a "one-prompt-for-all" strategy, where the model autonomously devises a step template (algorithm) for each task. Once the template is established, these approaches help navigate the answer space more effectively. For instance, Graph-of-Thought encourages the model to frequently revisit previously generated steps, while Tree-of-Thought allows the model to generate multiple possible next steps before selecting the most promising one. However, the step template itself (algorithm) is still generated by the model and can be poorly suited to the problem (Figure 1.a), especially when task-specific supervision (guidance) is lacking.

In this work, we thoroughly investigate the distinction between prompt space and answer space in the CoT process. Building on insights from previous theoretical analyses of CoT (Li et al., 2024), we explore why "supervision" is necessary and how it can be provided to guide the model in finding the optimal steps. We conduct extensive experiments on structured reasoning tasks, demonstrating that task-specific "supervised" CoT is crucial for achieving optimal solutions and highlighting the performance gap when supervision is used versus when it is not. Our work is the first of its kind to focus on prompt space exploration and offers valuable insights into *understanding* and *designing* effective prompt techniques for reasoning tasks.

## 2 DEMYSTIFYING CoT: A STRAIGHTFORWARD UNDERSTANDING

In this section, we summarize key findings from previous theoretical analyses (Li et al., 2024; Zhang et al., 2024; Feng et al., 2024) of CoT prompting, presenting them in a unified and accessible manner. The conclusions drawn here will serve as a foundation for our subsequent analysis of supervised CoT.

### 2.1 LIMITATIONS OF TRANSFORMER ARCHITECTURE

Transformers, unlike recurrent networks, are not designed to perform reasoning over an arbitrary number of sequential steps (depth) internally. Specifically, in a Transformer model, the hidden state $\mathbf{h}_{t-1}$ at time step $t-1$ is not reused when calculating $\mathbf{h}_t$ (Figure 2.b), as it would be in recurrent networks like RNN (Figure 2.a). Instead, the hidden state $\mathbf{h}$ is passed forward only through the *layers* of the Transformer (Dehghani et al., 2018) (Figure 1.c), not through time, which means that the number of sequential steps is fixed and limited for any given Transformer architecture (Li et al., 2024; Zhang et al., 2024; Elbayad et al., 2019). In contrast, Recurrent Neural Networks (RNNs) (Grossberg, 2013) allow the hidden state $\mathbf{h}$ to be passed through time steps via recurrent connections (Figure 2.a), enabling sequential computation over $\mathbf{h}$ through an arbitrary number of

input tokens. This capability allows RNNs to perform deeper reasoning over $\mathbf{h}$, which is essential for solving complex tasks (Zhang et al., 2024).

The hidden state $\mathbf{h}$ plays a crucial role in reasoning, as it stores both reasoning memory and intermediate reasoning results (Zhang et al., 2024). The ability to sequentially compute and update $\mathbf{h}$ over time allows a model to build reasoning depth, which is necessary for addressing complex problems. This depth advantage provided by recurrent connections cannot be replicated by autoregressive models. Autoregressive models, instead of passing the hidden state $\mathbf{h}_t$ forward, pass the generated token $\mathbf{y}_t$. However, $\mathbf{y}$ cannot replace the role of $\mathbf{h}$ for the following reasons: $\mathbf{y}$ is a discrete value extracted from $\mathbf{h}$ and only contains partial information (Figure 2.b), making it insufficient for continued reasoning in many tasks. $\mathbf{y}$ exists outside the latent space where $\mathbf{h}$ operates (Figure 2.b), meaning it cannot be used for computation in the same way that $\mathbf{h}$ can (Zhang et al., 2024).

## 2.2 NATURE OF REASONING

Reasoning inherently requires *sequential* depth. For tasks with input of length $\mathbf{n}$, reasoning is typically performed step by step to arrive at the final result. Examples include counting (incrementing a counter iteratively), playing chess (updating the board state iteratively), and searching (marking visited nodes iteratively). To solve a given task, there is a theoretical lower bound on the required depth of computation (Sanford et al., 2024). Since models like Transformers can only perform a constant number of sequential reasoning steps over the hidden state $\mathbf{h}$, they are unable to solve reasoning tasks where the depth requirement increases with the length of the input.

Consider chess as an example. For a sequence of chess moves, $\mathbf{x}_n = (\mathbf{x}_1, \mathbf{x}_2, \ldots, \mathbf{x}_n)$, to validate the $\mathbf{n}$-th move, the $\mathbf{n}$-th board state $\mathbf{h}_n$ must be calculated. This requires $\mathbf{n}$ *sequential* computations, as the $\mathbf{n}$-th board state depends not only on the sequence of moves $\mathbf{x}$ but also on the previous board state $\mathbf{h}_{n-1}$. While a neural network could *memorize* the mapping from $\mathbf{x}_n$ to the correct $\mathbf{h}$ (Arpit et al., 2017), bypassing the need for sequential computation, memorization is much more resource-intensive than reasoning. This is because memorization would require storing all possible *permutations* of $\mathbf{x}_n$ and their corresponding resulted board states, an exponential challenge that eventually demands infinite memory to store instances of arbitrary length.

Thus, in the example of simulating a chess game, the model's internal representation $\mathbf{h}$, which encodes the board state, must be sequentially computed $\mathbf{n}$ times to simulate the game. Transformers, which lack the infinite precision needed for memorization, cannot perform such tasks, as their hidden states $\mathbf{h}$ are computed a fixed number of times, regardless of the input length.

## 2.3 CoT + AUTOREGRESSIVE = RECURRENT

As previous studies have shown Li et al. (2024); Zhang et al. (2024); Feng et al. (2024), Chain of Thought (CoT) effectively bridges the gap between autoregressive Liang et al. (2022); Liu et al. (2022) models and recurrent structures Zhang et al. (2024) within large language models (LLMs). Instead of merely outputting tokens to answer questions, CoT also generates intermediate *steps* which are not part of the answers. These intermediate steps, represented as a sequence of natural language tokens $(\mathbf{o}_1, \mathbf{o}_2, \ldots, \mathbf{o}_k)$, act as a discretization of the latent information $\mathbf{h}_n$ (Figure 2.c). Given that natural language is a powerful medium for encoding nearly any type of information, $\mathbf{h}$ is effectively transformed into a token sequence $\mathbf{o}$, which is then converted back into a vector $\mathbf{h}$ via the embedding layer. In this way, computational information is preserved through a process of discretization followed by vectorization, represented as: $\mathbf{h}_t \xrightarrow{\text{discritization}} (\mathbf{o}_1, \mathbf{o}_2, \cdots, \mathbf{o}_k) \xrightarrow{\text{vectorization}} \mathbf{h}_{t+1}$ (Figure 2.c). This approach, effectively achieve the same effect as $\mathbf{h}_t \Rightarrow \mathbf{h}_{t+1}$ in the RNN-like recurrent network, allowing $\mathbf{h}$ to be recurrently updated by the network.

In the earlier chess example, the LLM generates intermediate reasoning steps as natural language strings during the CoT process. Specifically, it produces a sequence of tokens (e.g., in English) to describe the board state $\mathbf{h}_k$ after the first $\mathbf{k}$ moves, detailing the positions of pieces such as the bishop and the king. In the subsequent computation, the LLM reads this board description up to move $\mathbf{k}$ and uses it to calculate the $\mathbf{k+1}$-th board state, thereby avoiding the need to re-compute the reasoning from scratch—something Transformers cannot do internally due to their non-recurrent architecture.

In conclusion, LLMs with CoT effectively extend the reasoning process from the model's internal latent space $\mathbb{H}$ to a natural language-based token space $\mathbb{O}$. Thanks to the powerful encoding ability of natural language, intermediate reasoning steps are encoded and stored in text form, which the model can reuse in subsequent computations. This approach significantly increases the model's reasoning depth to $T(\mathrm{n})$, where $T(\mathrm{n})$ is the number of CoT steps performed. Under ideal theoretical conditions—such as infinite CoT steps and perfect information conversion between latent and text space—LLMs with CoT can achieve Turing completeness, theoretically solving any problem, including those beyond symbolic tasks (e.g. recognizing regular languages). This theoretical analysis provides strong guidance for designing effective "supervised" CoT approaches, which we introduce in subsequent sections.

## 3 CoT Search Space = Prompt Space + Answer Space

While theory suggests CoT-augmented LLMs can solve any problem Li et al. (2024), finding solutions in practice is much harder. CoT is limited by a finite number of steps, and the conversion from latent states $\mathbf{h}$ to token sequences $\mathbf{o}$ is imperfect. Consequently, only partial information is extracted at each step, making it crucial to identify the right data to continue the correct computation. We decompose the CoT reasoning into two components: template search within the prompt space and answer search within the answer space. We show how effective navigation of the prompt space can simplify answer space complexity and reveal limitations of unsupervised "X-of-thought" methods.

### 3.1 Prompt Space

The latent vector $\mathbf{h}$ contains rich intermediate information when processing a task, including counters, sums, flags for binary indicators, and more. When LLMs are prompted to "think step by step" along with the task instance, they generate a *step template*, specifying which information from $\mathbf{h}$ to extract and discretize into tokens $(\mathrm{o}_1, \mathrm{o}_2, \ldots, \mathrm{o}_k)$. Ideally, as $\mathrm{k} \to \infty$—meaning the length of the CoT is arbitrarily long—all vectorized information in $\mathbf{h}$ can be fully textualized, achieving *true* recurrence through autoregression. However, with limited $\mathrm{k}$, only partial information is discretized.

If we define the amount of information stored in $\mathbf{h}$ as $\mathrm{m}$ bits, and each CoT step extracts up to $\mathrm{s}$ bits of information into $\mathbf{o}$, each unique *step template* specifies a way to extract $\mathrm{s}$ bits from the full $\mathrm{m}$-bit space. Thus, the total number of potential step templates is $C(\mathrm{m}, \mathrm{s}) = \frac{\mathrm{m}!}{\mathrm{s}!(\mathrm{m}-\mathrm{s})!}$, which *estimates* the number of ways information can be extracted via CoT at each step. Each template defines an extraction of unique $\mathrm{s}$ bits of information.

For example, in the chess simulation case, $\mathbf{h}$ encodes details such as the `<current board layout>`, `<the next player>`, `<board status>`, `<number of pieces taken by each player>` and so on. When given the instruction to "think step by step", the model decides which information to extract based on the *step template* it generates. Extracting the wrong information might hinder reasoning in subsequent steps as *recurrence* can not be effectively performed on the needed information.

The prompt search complexity $C(\mathrm{m}, \mathrm{s})$ depends on both $\mathrm{m}$, the total information in $\mathbf{h}$, and $\mathrm{s}$, the amount of information each CoT step can extract. If a model is *sufficiently trained*, the total amount of encoded information in $\mathbf{h}$ is proportional to the dimension size of $\mathbf{h}$ (Allen-Zhu & Li, 2023), $\mathrm{d}$, denoted by $\mathrm{m} \propto \mathrm{d}$. In this context, $\mathrm{m}$ represents the size of the search space, while $\mathrm{s}$ correlates with the length of CoT tokens $\mathbf{o}$, as longer CoT steps tend to extract more information from $\mathbf{h}$. Thus, $\mathrm{s}$ serves as the search step size. In practice, step template search is not entirely random. Models often find relevant templates using heuristics, which significantly reduces the search complexity of $C(\mathrm{m}, \mathrm{s})$. However, identifying the optimal template remains challenging, and using an incorrect template can severely degrade performance, as demonstrated in our experiments.

In conclusion, the *step (prompt) template* defines how information is extracted and used *recurrently* in the CoT process. Finding the correct template is equivalent to discovering the *algorithm* for solving a given task, determining what information is needed at each step and how it should be used to compute the next state.

## 3.2 Answer Space

Once the model "decides" on the steps to follow during CoT, it performs reasoning accordingly. With a specific step (prompt) template $\mathrm{p}i$ chosen from the prompt space $\mathcal{P}$, CoT iteratively executes $\mathbf{h}_t \xrightarrow{\mathrm{p}_i} (\mathrm{o}_1^{(i)}, \mathrm{o}_2^{(i)}, \ldots, \mathrm{o}_k^{(i)}) \Rightarrow \mathbf{h}_{t+1}$ to update $\mathbf{h}$ and calculate the next state, continuing this process until reaching the final state (solution). The complexity of finding solutions in the answer space depends on both the choice of $\mathrm{p}_i$ and the nature of the task itself.

Each task embeds a different level of complexity in its answer space. For instance, in the chess simulation task of <finding a set of actions leading to game end>, the answer space $\mathcal{S} = (\mathbf{s}_1, \mathbf{s}_2, \ldots, \mathbf{s}_\infty)$ contains all possible combinations of action sequences $\mathbf{s}$. The solution set $\mathcal{CR} \subset \mathcal{S}$ includes all valid action sequences that lead to the end of the game, being a subset of the entire answer space $\mathcal{S}$. Solving the problem requires identifying one single correct action sequence $\mathbf{s}_{\mathrm{correct}} = (\mathrm{y}_1, \mathrm{y}_2, \ldots, \mathrm{y}_T) \in \mathcal{CR}$.

If a fixed step (prompt) template for this task, such as $\mathrm{p}_0$ = <extract current board configuration at each step>, is used, the CoT process iteratively extracts the current board description and use it for calculating next board state in $\mathbf{h}$ to identify the valid next move $\mathrm{y}_i$, eventually forming the correct answer $\mathbf{s}_{\mathrm{correct}} = (\mathrm{y}_1, \mathrm{y}_2, \ldots, \mathrm{y}_T)$. The complexity of navigating the answer space can be roughly measured by:

$$\frac{\mathrm{len}(\mathcal{CR})}{\mathrm{len}(\mathcal{S})} \quad | \quad \mathrm{p} \tag{1}$$

This ratio measures the proportion of the solution space $\mathcal{CR}$ relative to the entire answer space $\mathcal{S}$, *given* a specific template $\mathrm{p}$. If the chosen template $\mathrm{p}$ extracts irrelevant information—such as determining which player is next at each step—the ratio simplifies to $\frac{\mathrm{len}(\mathcal{CR})}{\mathrm{len}(\mathcal{S})}$. In this case, each $\mathrm{y}_i$ would be generated randomly, as $\mathbf{h}$ can not be computed iteratively over useful information needed for extracting correct $\mathrm{y}_t exttti$, making the correct answer only discoverable by *chance*.

Correctly identifying the step template $\mathrm{p}$ is crucial for reducing the complexity of $\frac{\mathrm{len}(\mathcal{CR})}{\mathrm{len}(\mathcal{S})} \mid \mathrm{p}$, as $\mathrm{p}$ dictates what information is recurrently overlayed in the process $\mathbf{h}_t \Rightarrow \mathbf{h}_{t+1}$ and in turn what can be calculated, essentially acting as the "algorithm" for solving tasks in the CoT process. In the chess example, the optimal template would be <extract current board configuration at each step>, allowing the model to reason over the board state iteratively, i.e., $\mathbf{h}_t \xrightarrow{\text{board state}} \mathbf{h}_{t+1}$. With the correct board state computed recurrently, the valid next move $\mathrm{y}_t$ can be effortlessly derived from $\mathbf{h}_t$. However, using a less relevant template, such as <extract the number of pieces on the board at each step>, would expand the search space nearly to $\frac{\mathrm{len}(\mathcal{CR})}{\mathrm{len}(\mathcal{S})}$, as the number of pieces doesn't provide useful information for determining the next valid move. Consequently, the model would have to recalculate the board state at each step from previously generated moves $\mathrm{y}_1$ , which requires $O(n)$ depth–Transformers, limited by constant depth, cannot handle. As a result, the next action $\mathrm{y}t+1$ would not benefit from the CoT process.

## 3.3 CoT as an **Unsupervised** Task Solver

CoT operates in an unsupervised manner for any given task, relying on a single universal prompt, Think Step by Step, and leaving it to the model to generate its own step template $\mathrm{p} \in \mathcal{P}$ for extracting information at each step. Since humans do not supervise step completion, the generation of steps—i.e., determining which information to extract from $\mathbf{h}$ and compute recurrently—comes primarily from the model's heuristics. For example, in counting tasks, LLMs use learned heuristics to extract a *Counter* value from $\mathbf{h}$ and perform recurrent updates. However, these unsupervised, heuristic-driven templates are often unreliable, as the model lacks the knowledge to identify key components for computation, as demonstrated in previous work Valmeekam et al. (2022) and our experiments.

### 3.4 CoT Variants as Unsupervised Helpers for Navigating Answer Space

In practice, the answer space $\mathcal{S}$ can be large and complex, and even with the optimal step (prompt) template p, CoT can make errors. Various CoT variants, such as Tree-of-Thought (ToT) and Graph-of-Thought (GoT), have been proposed to mitigate these mistakes in solution searching. While these "X-of-thought" approaches don't dictate which specific information to extract at each step like p does, they improve solution finding by exploring multiple paths and self-verifying. For instance, ToT explores multiple instances in the answer space simultaneously under some given template p, unlike the single-path exploration of CoT. Specifically, information extracted from the current hidden state $\mathbf{h}_t$ using p is used to generate q possible answers for the next step, denoted as $(\mathbf{y}_{t+1}^{(1)}, \mathbf{y}_{t+1}^{(2)}, \ldots, \mathbf{y}_{t+1}^{(q)})$. Each answer leads to a different next state $\mathbf{h}_{t+1}$. In the example of `<finding a set of actions leading to game end>`, the board state at step t is extracted into descriptions using the correct template p and to form $\mathbf{h}_{t+1}$, and instead of producing a single next move $\mathbf{y}_{t+1}$ from $\mathbf{h}$, multiple actions are derived. Each derived action along with previous actions forms a unique path that leads to a potential solution in $\mathcal{S}$. Since some paths may fail (e.g., leading to a non-ending game), exploring multiple paths simultaneously increases the efficiency of searching the answer space. The visualization is shown in Figure 3.

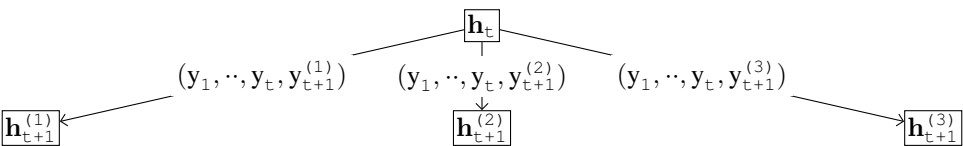

Figure 3: ToT mechanism. $\mathbf{h}_t$ is transitioned into different $\mathbf{h}_{t+1}$, to explore more in *answer space*. How state is transitioned is dictated by the step template of CoT, which goes beyond what ToT offers.

Similarly, GoT improves search accuracy by iteratively revisiting previously generated *partial answers*. However, none of these approaches are supervised, as the model is not informed of the correct step template p and generates it on its own, extracting information at each step accordingly. X-of-Thought still relies on a "one-prompt-for-all" approach and only aids in finding answers after $p \in \mathcal{P}$ is fixed. As we have shown, this can lead to poor outcomes, since p directly influences the complexity of the answer space, and X-of-Thought may be too late to correct errors in some cases.

## 4 Experiments

In this section, we conduct experiments to demonstrate the importance of supervision in the CoT process. Specifically, we design scenarios where the correct step template is provided through supervision, and compare them to cases where incorrect steps are simulated by the model. Our results show significant performance degradation when the step templates are incorrectly derived, highlighting the need for human supervision to ensure reliable task performance with LLMs.

*The objective of our experiments is not to evaluate the reasoning performance of different LLMs, but to emphasize the critical role that "supervision" plays in CoT. Comparing the abilities of various models is beyond the scope of this work.*

### 4.1 Experiments Designs

Although we used chess simulation as an example of reasoning with CoT due to its resemblance to real-life complex reasoning tasks, tasks involving chess boards and actions can be difficult to implement and evaluate. Instead, we follow previous work Zhang et al. (2024); Delétang et al. (2022) by focusing on more fundamental reasoning tasks for LLMs. Specifically, we evaluate tasks at three levels of computability: Regular (R), Context-Free (CF), and Context-Sensitive (CS), each corresponding to tasks solvable by different levels of computational power, from deterministic automata all the way to linear bounded automata (restricted Turing machines). These tasks involve operations such as counting, sorting, and number addition—basic operations that are required by more complex algorithmic problems (like NP problems). Each task has a strong dependency on identifying

| Level | Task | RNN | Tape RNN | Transformer | LLM w/o CoT | CoT Unsupervised | CoT CR Supervised | CoT IN Supervised |
|---|---|---|---|---|---|---|---|---|
| R | Modular Arithmetic | **1.00** | **1.00** | 0.24 | 0.22 | **0.96** | **1.00** | 0.44 |
|  | Parity Check | **1.00** | **1.00** | 0.52 | 0.58 | **0.94** | **1.00** | 0.42 |
|  | Cycle Navigation | **1.00** | **1.00** | 0.62 | 0.50 | 0.78 | **1.00** | 0.26 |
| CF | Stack Manipulation | 0.56 | **1.00** | 0.58 | 0.00 | **0.92** | **0.96** | 0.00 |
|  | Reverse List | 0.62 | **1.00** | 0.62 | 0.00 | 0.80 | **0.96** | 0.38 |
|  | Modular Arithmetic | 0.41 | **0.95** | 0.32 | 0.00 | 0.82 | **0.94** | 0.50 |
| CS | Odds First | 0.51 | **1.00** | 0.53 | 0.00 | 0.80 | **0.92** | 0.00 |
|  | Addition | 0.50 | **1.00** | 0.54 | 0.00 | 0.84 | 0.88 | 0.00 |
|  | Multiplication | 0.50 | 0.59 | 0.52 | 0.00 | 0.14 | 0.44 | 0.00 |
|  | Sorting | 0.28 | 0.71 | **0.92** | 0.00 | 0.36 | **0.90** | 0.00 |

Table 1: Main results across three levels of reasoning tasks. For LLMs without CoT, intermediate steps are explicitly prohibited in the prompt. In CoT generation, "CR Supervised" refers to when we provide the correct supervision. "IN Supervised" refers to when seemingly correct but suboptimal step templates are provided, simulating scenarios where the model makes mistakes in navigating the prompt space and derives incorrect step templates. **Bolded** numbers indicate performance greater than or equal to 0.9, while red indicates low performance (below 0.2). Results for RNN, Tape-RNN and Transformer are trained expert model by previous research (Delétang et al., 2022), they are solely used for reference and not compared with LLMs as it follows slightly different experiment settings.

| Model | R | | | CF | | | CS | | | |
|---|---|---|---|---|---|---|---|---|---|---|
|  | MA | PC | CN | SM | RL | MA | OF | AD | MU | SO |
| Unsupervised **CoT** | **0.96** | **0.94** | 0.78 | 0.92 | 0.80 | 0.82 | 0.80 | 0.84 | 0.14 | 0.36 |
| Unsupervised **ToT** | **0.92** | **0.90** | **0.92** | 0.36 | 0.88 | 0.78 | 0.82 | **0.94** | 0.18 | 0.66 |
| Unsupervised **GoT** | **1.00** | **0.98** | **0.90** | 0.72 | **0.92** | 0.88 | 0.82 | **0.92** | 0.20 | 0.80 |
| Correctly supervised **CoT** | **1.00** | **1.00** | **1.00** | 0.96 | 0.96 | 0.94 | 0.92 | 0.88 | 0.44 | 0.90 |

Table 2: Variant of CoT in performing each task. Each task is named using the first two letters in Table 1.

the correct step template, thus allowing us to clearly observe the impact of selection on step template on CoT performance.

All of these tasks require a level of computability beyond the capabilities of the Transformer's internal architecture Delétang et al. (2022). Specifically, they demand a *minimum* computational depth that scales linearly with input length, surpassing the constant depth inherent to Transformer models. Thus, solving these tasks necessitates the use of CoT, and correctly identifying the information to extract during CoT is crucial for resuming computation and building the necessary depth.

We use GPT-4-o classic, a version that eliminates the use of external tools (e.g., calculators or programs) and functions solely based on the model itself. We test each task using instances sampled according to previous work (Zhang et al., 2024). To ensure that factors such as long-context information retrieval and tokenization do not affect the results, we follow the setup from prior research and conduct controlled experiments. Details of our experimental design, including length sampling, task specifications, format adjustments, and prompt usage, are provided in the Appendix.

We extend the previous findings on expert models Delétang et al. (2022), which are specifically trained for particular tasks, to our experiments with LLMs. Due to differences in experimental settings, the results from expert models are presented for reference rather than direct comparison. Unlike prior research, which reports the best performance out of $N$ trials Delétang et al. (2022); Zhang et al. (2024) for each task instance, we report the average one-trail performance across all tested instances. Our focus is on practical usability beyond the theoretical upper-bound computability analysis in previous work. The final results are shown in Table 1.

## 4.2 MAIN RESULT

**Recurrence is key for reasoning.** As demonstrated in both expert models (RNN, Tape-RNN, and Transformers) and LLMs, recurrence is the determining factor for *solving* tasks in each category.

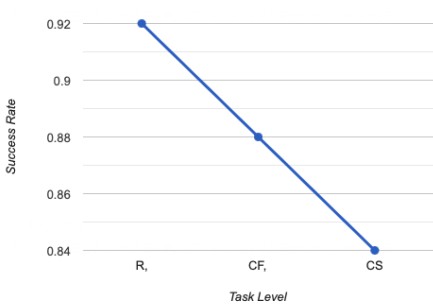

Figure 4: Average success rate in deriving correct step template in each level of tasks.

Specifically, expert models like RNN and Tape-RNN show the ability to solve tasks across various categories with over 90% accuracy, depending on their memory architecture. Transformers, however, are limited by their shallow depth of reasoning, as shown earlier, and fail to solve any tasks. Similarly, LLMs without CoT, relying solely on internal Transformer reasoning, achieved 0% performance on most tasks, with low performance on others likely due to guessing. When CoT augments LLMs with recurrent computational power, accuracy improves significantly. These comparisons highlight the critical role of *recurrence* in a model's computability, reinforcing the analysis we previously discussed.

**Role of Step Template in Reasoning Performance: Supervision Is Essential.** We provide human supervision for all tasks, and we observed that, due to the relatively simple nature of the tasks, the model makes mistakes in finding the *optimal* step template less frequently. As a result, it is difficult to clearly observe the performance gap between optimal and non-optimal step templates. To address this, we introduce two types of supervision for each task: Correct Supervision (CR Supervised), where the model is guided with optimal steps to demonstrate the best possible performance, and Incorrect Supervision (IN Supervised), which simulates scenarios where the model derives incorrect steps to show how performance can degrade. We present examples of these supervised scenarios for each task in Table 3.

| Task | Correct Supervision Example | Incorrect Supervision Example |
|---|---|---|
| **Modular Arithmetic** | Write down partial sums after each step | Write down paired sums of each two values at each step |
| **Parity Check** | Write down "even" or "odd" counter after each word in each step | Write down whether the word is a target word at each step |
| **Cycle Navigation** | Write down which state you are in at each step | Write down the total number of "forward" at each step |
| **Stack Manipulation** | Write down the resulting stack at each step | Write down the number of operations performed up to that step |
| **Reverse List** | Write down the partially reversed list after each step from the back | Write down the value to be added to the reversed list and the remaining original list |
| **Modular Arithmetic** | Write down the formula with reduced values in the performed operations at each step | Write down the result of each performed operation at each step |

Table 3: Examples of correct and incorrect steps for performed reasoning tasks.

From Table 1, we observe that providing supervision yields noticeable improvements over the unsupervised "step-by-step" approach. Specifically, errors caused by the model's own derived step templates are eliminated with correct supervision, resulting in better performance scores. In contrast, when the step template is intentionally set up incorrectly, we observe a *significant* performance degradation, with some tasks performing as poorly as they would without using CoT.

To explain this further, when a step template is incorrectly specified (e.g., outputting the sum up to the current step for a task that requires counting appearances), the useful counter information $c$ in $h_t$ is not extracted. As a result, $c$ is not carried forward into the next state $h_{t+1}$, leading to a failure in resuming the necessary calculations. While the wrongly specified information (e.g., the partial sum) is recurrently calculated, it does not lead to the correct final answer for the task.

**CoT Variants are Useful in Navigating Answer Space.** We compare the results of different CoT variants for the same tasks. As shown in Table 2, both ToT and GoT improve performance over naive CoT. However, this improvement is due to correcting "incorrect calculations" during computation, not from improvements in step-template selection. ToT provides little benefit, as the tasks typically have only one path to the solution. In contrast, GoT shows greater accuracy gains, thanks to its self-revisiting mechanism,

**Prompt Space Analysis.** We further analyzed the model's performance in navigating the prompt space, i.e., finding the correct (optimal) step template for each task. As shown in Figure 4, all tasks involve relatively simple calculations, and the model exhibits a high average success rate in identifying the correct template. Specifically, the success rate for R-type tasks exceeds 90%. As task complexity increases, we observe a slight decline, with CS tasks showing an 84% success rate in extracting the correct information during CoT. We further include case studies showcasing how "sub-optimal" steps are derived from unsupervised CoT process, which are shown in Figure 1 and Appendix Figure 5, 6 and 7.

Lastly, we showcase how incorrect navigation in the prompt space leads to uncorrectable results. As shown in Appendix Figures 8 and 9, the incorrect step template results in incorrect information extraction, leading to a wrongly computed next state and ultimately increasing the difficulty of searching the answer space.

# 5 SUPERVISED CoT: USERS' PERSPECTIVE

## 5.1 HOW TO SUPERVISE?

As we've demonstrated, providing correct supervision is crucial for helping the model achieve accurate results. A natural question arises: how can effective supervision be derived? The key to good supervision lies in understanding CoT's underlying mechanism, which essentially involves relaying information through the text space. For tasks requiring multiple steps, users need to identify *what each step is* and *what key information should be extracted at each step*. While this might seem straightforward in the basic reasoning tasks used in our experiments, it becomes more complex for challenging tasks, where correctly identifying the information requires careful task analysis. Therefore, human knowledge is critical for enhancing the model's computational abilities and can directly influence task success. However, this supervision adds substantial workload, as each task demands a unique understanding of its computational structure.

## 5.2 WHEN TO SUPERVISE?

As we've observed, using an incorrect step template—whether model-derived or human-injected—can result in significant performance degradation. Based on this, it's important to avoid providing supervision unless you are reasonably confident that the steps will not hinder the reasoning process. In cases of uncertainty, it may be better to rely on the model's own heuristics.

# 6 CONCLUSIONS

Our work offers a unique perspective on the mechanics of Chain of Thought (CoT) prompting and its role in enhancing model reasoning. Through theoretical analysis and practical insights, we show how CoT transforms latent information into text space, enabling iterative and resumable reasoning steps that expand a model's computational depth. We further connect the model's problem-solving capabilities with the complexities of finding solutions. Our analysis of prompt space and answer space underscores the importance of identifying the correct step template to simplify navigation—an often overlooked aspect in prompt-related research. The success of CoT hinges not only on generating steps but on extracting the right information at each stage. Our experiments demonstrate that incorrect step templates can severely impact reasoning, reinforcing the importance of supervision. Even small errors in template selection can lead to significant failures. Our findings combine theoretical analysis and experimental evidence, offering valuable insights into CoT's limitations and potential for improving reasoning tasks in large language models.

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

# A APPENDIX

## A.1 EXPERIMENTAL DESIGN

Our experimental setup carefully addresses potential pitfalls that could influence the model's performance, specifically focusing on tokenization and context length. Tokenization issues can significantly affect how models handle specific tasks, often leading to failures not tied to the model's reasoning ability. To counter this, we reformatted task instances to eliminate tokenization biases. Moreover, LLMs often struggle with retrieving information from long contexts, leading to hallucinations or forgotten data during extended reasoning processes. This tends to degrade accuracy, as models fail to maintain accurate references to the initial task elements throughout longer sequences. While these challenges are important in real-world applications of LLMs, they are outside

the scope of our investigation, which prioritizes analyzing the effect of using different step template. To maintain a controlled environment, we restrict task lengths from 20 to 35 elements. This threshold was determined from preliminary analysis, where longer task sequences often introduced issues not related to reasoning but to the model's internal optimization process. When the task sequence exceeds 35 steps, models can divide output over multiple contexts, which distorts accurate information retrieval. By maintaining a manageable length, we isolate and evaluate the differences between reasoning with and without CoT, avoiding disruptions caused by excessive context length. For each task, we generate 50 instances using a pre-written script and the results are examined by humans.

TASK DESIGN

Each task involves simple rule-based iterations, emphasizing memory access and iterative processes. The challenge for the model lies in its ability to execute these tasks within the constraints of its architecture and memory systems. Below, we describe each task in detail, including sample inputs and outputs. For the Regular (R) class tasks, we include the following:

TASK DESIGN

For the Regular (R) class tasks, we include the following:

TASK DESIGN

For the Regular (R) class tasks, we include the following:

1. **Modular Arithmetic:** Given a sequence of $n$ numbers and basic operations (+, -), compute the result modulo 5. For example, the input $4 + 2 - 3$ should yield 3.

2. **Parity Check:** Determine if the word "banana" appears an even number of times in a list containing the words "apple" and "banana." For example, the input (`"banana"`, `"apple"`, `"banana"`) yields `True`.

3. **Cycle Navigation:** Based on a sequence of actions ("forward," "backward," "stay"), determine the final position in a 5-state cycle starting from state 1. For example, (`"forward"`, `"stay"`, `"backward"`) will return state 1.

For the Context-Free (CF) class tasks, we use the following:

1. **Stack Manipulation:** Given a list of fruit names representing a stack and a sequence of stack operations, compute the final stack. For example, applying (pop `"banana"`, push `"orange"`) to (`"apple"`, `"banana"`, `"grape"`) results in (`"apple"`, `"orange"`, `"grape"`).

2. **Reverse List:** Reverse a list of vegetable names. For example, (`"carrot"`, `"potato"`, `"onion"`) becomes (`"onion"`, `"potato"`, `"carrot"`).

3. **Modular Arithmetic (Complex):** Compute the result of an arithmetic expression with $n$ operations modulo 5. For instance, $((2 + 4) \times (3 - 1)) \mod 5$ yields 0.

For the Context-Sensitive (CS) class tasks, we evaluate the following:

1. **Odd First:** Extract all items at odd positions from a list of animal names, followed by those at even positions. For example, (`"dog"`, `"cat"`, `"elephant"`, `"tiger"`) yields (`"dog"`, `"elephant"`, `"cat"`, `"tiger"`).

2. **Addition:** Given two large numbers with $n$ digits, calculate their sum. For instance, the input $123456 + 987654$ yields 1,111,110.

3. **Multiplication:** Multiply two large numbers with $n$ digits. For instance, the input $345 \times 567$ yields 195,615.

4. **Sorting:** Sort a list of integers using the insertion sort algorithm. For example, the input (8, 3, 5, 1) would result in (1, 3, 5, 8).

> reverse this list, think step by step: ['dog', 'cat', 'cat', 'banana', 'apple', 'fruit', 'fruit', 'meat', 'apple', 'banana', 'cat', 'apple', 'banana', 'city', 'dot', 'city', 'meat', 'apple', 'dog', 'fruit', 'cat', 'dot', 'apple', 'meat', 'meat', 'apple', 'cat', 'fruit', 'banana', 'meat', 'apple', 'apple', 'banana', 'cat', 'city', 'fruit', 'cat', 'city', 'dog', 'dot']

To reverse the list step by step, we take each element from the end and move it to the beginning. Let's go through the process.

The original list is:

```python
['dog', 'cat', 'cat', 'banana', 'apple', 'fruit', 'fruit', 'meat', 'apple', 'ban
```

**Step-by-Step Reversal:**

- The last element is `'dot'`, move it to the first position.
- Next is `'dog'`, place it after `'dot'`.
- Then `'city'`, place it after `'dog'`.
- Then `'cat'`, place it after `'city'`.

Figure 5: The model-derived template, <extract next move>, is sub-optimal and leads to incorrect results. The resulting partial list is not recurrently generated and computed, and the final list is only generated at the end. Note that the full output is not shown due to its length.

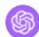 calculate the new list with values at odd positions followed by even positions in the following list: ['tree', 'river', 'car', 'dog', 'sun', 'moon', 'apple', 'cat', 'dog', 'car', 'city', 'fruit', 'banana', 'river', 'dot', 'moon', 'meat', 'fruit', 'apple', 'dot', 'cat', 'banana', 'tree', 'sun', 'city', 'apple', 'meat', 'dog', 'fruit', 'tree', 'banana', 'dot', 'cat', 'meat', 'river', 'moon', 'city', 'dog', 'fruit', 'apple']. Do this step by step.

Let's break down the task step by step.

**Step 1: Identify Odd and Even Positions**

In Python (or most programming languages), list indices start at 0. Therefore:

- Odd positions (1-based) correspond to even indices (0-based).
- Even positions (1-based) correspond to odd indices (0-based).

**Step 2: List Elements at Odd Positions (0-based)**

These correspond to even indices (0, 2, 4, etc.).

Let's extract those elements:

- Index 0: 'tree'
- Index 2: 'car'
- Index 4: 'sun'
- Index 6: 'apple'
- Index 8: 'dog'
- Index 10: 'city'
- Index 12: 'banana'
- Index 14: 'dot'
- Index 16: 'meat'
- Index 18: 'apple'
- Index 20: 'cat'
- Index 22: 'tree'
- Index 24: 'apple'                                  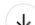

Figure 6: Model-derived template: `<write down each element at next position>`, which is sub-optimal and leads to incorrect results. The partial list is not computed recurrently during the step, and the final list is only generated at the end. Note that the full output is not shown due to its length.

Figure 7: Model-derived template: `<extract whether to swap at each step in bubble sort>`, which is sub-optimal and leads to incorrect results. The partial sorted list is not computed recurrently. Note that the full output is omitted due to its length.

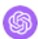

calculate -9 - 10 - 14 - 2 - 1 + 13 - 3 - 8 + 12 + 9 - 8 + 15 - 8 - 4 +
10 + 10 + 1 - 4 + 5 - 10 - 12 + 2 + 9 - 11 + 5 + 15 - 12 - 13 + 12 - 12
- 11 - 1 + 2 + 8 - 14 - 9 + 6 - 11 + 8 + 7 - 2, do it step by step, each
step i print the sum of 2i th value and 2i-1 th value

Let's break down the calculation step by step as per your request. Each step will add two
consecutive values from the list and print the result.

The list of numbers is:

$-9, -10, -14, -2, -1, 13, -3, -8, 12, 9, -8, 15, -8, -4, 10, 10, 1, -4, 5, -10, -12, 2, 9,$

Now let's add two consecutive numbers at a time:

1. Step 1: $-9 + (-10) = -19$
2. Step 2: $-14 + (-2) = -16$
3. Step 3: $-1 + 13 = 12$
4. Step 4: $-3 + (-8) = -11$
5. Step 5: $12 + 9 = 21$
6. Step 6: $-8 + 15 = 7$
7. Step 7: $-8 + (-4) = -12$
8. Step 8: $10 + 10 = 20$
9. Step 9: $1 + (-4) = -3$
10. Step 10: $5 + (-10) = -5$
11. Step 11: $-12 + 2 = -10$
12. Step 12: $9 + (-11) = -2$
13. Step 13: $5 + 15 = 20$
14. Step 14: $-12 + (-13) = -25$
15. Step 15: $12 + (-12) = 0$
16. Step 16: $-11 + (-1) = -12$
17. Step 17: $2 + 8 = 10$

Figure 8: Case study of performing simple arithmetic under a wrong template (intentionally provided through supervision). Since the cumulative sum is not derived iteratively, the intended value cannot be computed recurrently through CoT, leading to incorrect results. The correct step template for this task should be `<write down the calculated total value up to each step>`. Note that the full output is omitted due to its length.

given the stack ['meat', 'dot', 'cat', 'city', 'yellow', 'apple', 'meat', 'fruit', 'banana', 'book', 'dot', 'book', 'dog', 'city', 'banana', 'dot', 'cat', 'cat', 'meat', 'yellow', 'dot', 'book', 'dog', 'meat', 'cat', 'book', 'book', 'city', 'meat', 'dot', 'cat', 'book', 'city', 'meat', 'city', 'city', 'cat', 'book', 'yellow', 'meat'], calculate final list after operations: ['pop', 'pop', 'push "city"', 'pop', 'pop', 'push "dog"', 'pop', 'pop', 'pop', 'push "fruit"', 'push "dog"', 'push "fruit"', 'push "dot"', 'push "apple"', 'push "dog"', 'pop', 'push "city"', 'pop', 'push "cat"', 'push "meat"'], do it step by step, each step print total number of items

Let's go step-by-step through the stack with each operation and print the total number of items after each operation.

**Initial Stack:**

```plaintext
['meat', 'dot', 'cat', 'city', 'yellow', 'apple', 'meat', 'fruit', 'banana', 'bo
```

There are **39** items in the stack.

---

**Operation 1:** `'pop'`
- Remove the last element ( `'meat'` ).
- Stack now has **38** items.

**Operation 2:** `'pop'`
- Remove the last element ( `'yellow'` ).
- Stack now has **37** items.

**Operation 3:** `'push "city"'`
- Add `'city'` to the stack.
- Stack now has **38** items.

↓

**Operation 4:** `'pop'`
- Remove the last element ( `'city'` ).
- Stack now has **37** items.

**Operation 5:** `'pop'`
- Remove the last element ( `'book'` ).
- Stack now has **36** items.

**Operation 6:** `'push "dog"'`
- Add `'dog'` to the stack.
- Stack now has **37** items.

Figure 9: Case study of performing stack manipulation under a wrong template, yielding incorrect results (intentionally provided through supervision). Since the stack status is not iteratively updated and passed to the next state, the results cannot be tracked effectively. Tracking the total number of items in the stack is not useful for deriving the final stack. The correct step template for this task should be <write down the current stack status at each step>. Note that the full output is omitted due to its length.