# OpenReview forum: "Supervised Chain of Thought"
_ICLR.cc/2025/Conference — ICLR 2025 Conference Withdrawn Submission_

### Official Review · Reviewer_tXoh · 2024-11-03

**Soundness:** 1
**Presentation:** 2
**Contribution:** 1
**Rating:** 1
**Confidence:** 3

**Summary:**

The authors pose existing CoT prompting methods as an "unsupervised" method for reasoning in LLMs, and suggest that task-specific "supervision", i.e., CoT prompts that inform the model how to solve the given task, can enable more effective reasoning in LLMs.

The paper summarizes existing works on understanding CoT reasoning, some of which interpret LLMs equipped with CoT as recurrent networks, in which the hidden states are intermediately represented as a series of discrete states, i.e., reasoning tokens. Based on this formulation, the authors analyze the CoT search space in terms of the "prompt space" and "answer space". The authors suggest that standard CoT prompting relies on the model itself to navigate the prompt and answer space, with previous X-of-thought methods being unsupervised "helpers" to navigate the prompt space. The authors suggest that "supervised CoT", or prompts that outline the correct steps required to solved the task, can inform the model of the correct "prompt template" within the prompt space.

The authors evaluate task-specific RNNs and transformers, as well as LLMs with unsupervised and supervised CoT prompts (crafted by the authors) under three classes of tasks: regular (R), context-free (CF), and context-sensitive (CS). While task-specific models can solve all classes of tasks reliably, vanilla LLMs suffer in CF and CS. Unsupervised CoT improves performance but shows limitations in CS. Supervised CoT solves this limitation, and while incorrect supervision (misleading prompts) degrades performance.

Based on this, the authors conclude that recursion and "supervision" of the prompt space is essential for reasoning performance.

**Strengths:**

- The authors provide a comprehensive summary of previous works which aim to understand the mechanism of CoT reasoning
- The authors offer an interesting perspective on the role of CoT prompts

**Weaknesses:**

- **The conclusions are already well-established**: It is widely recognized that LLM task performance varies based on the given prompt, a fact that has driven the growth of prompt engineering as a specialized field. Similarly, it is not surprising that LLMs can demonstrate improved reasoning abilities when guided by prompts with structured reasoning steps.

- **Theoretical analysis offers limited insights and could benefit from further rigor**: The theoretical analysis in Section 3 provides an interesting analogy on the mechanism of CoT reasoning. However, some definitions and assumptions could be more precisely defined, and there is limited mathematical derivation that leads to meaningful conclusions. The main takeaway seems to be that both the prompt space (approach to the problem) and the answer space (execution of reasoning steps) are large. While this may serve as the motivation for the experiments, it does not provide theoretical support for its conclusions. Valid theoretical justification for a well-known phenomenon can be a meaningful conclusion, but further development of the analysis would be needed to provide such justification in this case.

*This section was written by the reviewer, re-worded using ChatGPT, and manually checked by the reviewer.*

**Questions:**

Please refer to weaknesses

---

### Official Review · Reviewer_4uM5 · 2024-11-03

**Soundness:** 3
**Presentation:** 2
**Contribution:** 2
**Rating:** 1
**Confidence:** 4

**Summary:**

This paper explores the concept of Chain of Thought in Large Language Models and emphasizes the importance of supervision for optimal reasoning performance. The Transformer-based architecture of most LLMs faces depth limitations, making it difficult to handle complex reasoning tasks. CoT has emerged as a strategy to overcome these limitations by guiding models through intermediate reasoning steps. However, conventional "unsupervised" CoT, where a single, generalized prompt is used, struggles with accuracy across varied tasks because it lacks tailored guidance for each task. The authors argue that dividing CoT into "prompt space" (step template selection) and "answer space" (iterative solution process) reveals the need for supervised CoT, where human-provided task-specific prompts significantly improve reasoning accuracy. The findings advocate for a more interactive CoT approach that requires human involvement in guiding complex tasks, providing insights into more effective CoT designs for improved reasoning depth and reliability in LLMs.

**Strengths:**

1. The paper is well-structured, with clear explanations of core concepts such as prompt space and answer space, and the step-by-step breakdown of task examples provides a clear understanding of the benefits of CoT.
2. While CoT itself is an established technique, this work takes a novel approach by critically analyzing the limitations of traditional, unsupervised CoT.

**Weaknesses:**

1. Although this paper discusses the importance of supervised CoT, this supervision is in the form of case-by-case human feedback. This paper does not draw a conclusion on how to make supervision based on the proposed theory.  While the paper demonstrates that task-specific supervision improves reasoning, it does not discuss the feasibility or cost of providing this supervision at scale. Implementing supervised CoT across various applications would likely demand substantial domain expertise and time.

2. The explanation of recurrent and autoregressive learning in section II of the article does not correspond to reality. In fact, the hidden state of models such as Transformer moves over the sequence in time in the form of a KV cache. Although I can understand that this form of hidden state is not reflective of global information with a limited context window, and that information beyond the window requires a memory to represent it. However the way the understanding is presented in the paper may be not a good reflection of the real cases.

3. The experiments focus on straightforward tasks, which limits the generalizability of the findings. Including more complex tasks with multi-step dependencies and nuanced, open-ended reasoning would strengthen the evidence for supervised CoT's broader applicability.

**Questions:**

1. Autoregressive models can also rely on large context windows, and in fact the vast majority of inference processes in LLM have intermediate steps that do not exceed the context window. So why do recurrent and autoregressive models differ in this paradigm when the only difference between them is implicit and explicit hidden state transfer?

2. Can the performance of LLM be improved by fine-tuning it for the dataset used?

3. Is it possible to give an automated, generalized supervision design methodology based on the described analysis of unsupervised CoT？

---

### Official Review · Reviewer_jx2n · 2024-11-04

**Soundness:** 3
**Presentation:** 3
**Contribution:** 2
**Rating:** 5
**Confidence:** 4

**Summary:**

This paper addresses an existing shortcoming in Chain-of-Thought (CoT) prompting, where a "one-prompt-for-all" approach is leveraged to generate reasoning steps. Previous research has shown that the transformer architecture has computational limitations in performing sequential computation. CoT has helped mitigate the "constant depth" limitation of transformers' internal reasoning.
The authors explain that all existing variants of "Chain-of-Thought" (such as vanilla chain of thought, Tree of Thought, and Graph of Thought) are unsupervised, meaning that the model generates its step template without task-specific supervision from humans. The unsupervised nature of these "X-of-Thought" methods results in underperformance. The authors reformulate the problem into "prompt space" and "answer space" and demonstrate how task-specific supervision is essential for accurately navigating the prompt space, simplifying the answer space, and achieving optimal performance. Through experiments with state-of-the-art LLMs, the authors reveal a significant gap in reasoning performance when supervision is applied versus when it is not.
They use GPT-4 to show how supervised CoT can significantly improve model performance across various tasks (such as sorting and parity checking) and can match the performance of RNNs and Tape-RNNs, which inherently solve tasks requiring deeper reasoning through their hidden states.

**Strengths:**

The authors address an important shortcoming of Transformers: unlike recurrent networks, they are not able to perform reasoning over an arbitrary number of sequential steps (depth). Since the number of sequential steps in transformers is fixed and limited by the number of layers, CoT provides a discretized approach to adding a hidden state in autoregressive transformers at each step. The authors then formulate CoT reasoning into 'prompt space' and 'answer space' and demonstrate the complexity of prompt space C(m,s) as a function of information (m bits) in hidden state h and information (s bits) in CoT step distcretized into token
The authors further show the relationship between the solution space CR and the entire answer space S, given a specific template p. They argue that selecting an appropriate prompt p (from the prompt space) is crucial for reducing the complexity of the answer space. Based on this argument, the authors contend that unsupervised CoT relies on heuristic-driven templates in the prompt space, which are often unreliable and result in a low chance of reaching the correct solution. To address this challenge, they propose using supervision to sample more informative prompt templates p from the prompt space, increasing the likelihood of discovering the correct solution in the answer space.

**Weaknesses:**

* **Unclear Focus on Discovery vs. Throughput**: While the paper's motivation is reasonable, it is unclear whether the authors aim to increase the chance of discovering the correct solution or the model's throughput (e.g., number of output tokens) by selecting the right prompt from the prompt space. In some examples, such as BFS versus DFS, different path lengths (number of output tokens) result in correct answers, but this is not clearly addressed.
* **Questionable Claim on Heuristic-Driven Templates**: The authors argue that prompt templates in unsupervised CoT come from heuristics. However, CoT often includes few-shot examples, so it is unclear why the authors believe the templates are heuristic-based rather than a result of the model’s generalization capabilities from the examples.
* **Lack of Clarity on Supervised Step Template Methodology**: It is unclear what techniques the authors use to provide "correct step templates" through supervision. They do not explain how these templates are validated (e.g., prompt engineering on a validation set) or why a language model cannot deduce the step template from a few-shot example.
* **Lack of Practical Framework**: The authors argue that selecting the correct "step template" (e.g., "Write down partial sums after each step") becomes more complex for challenging tasks. However, they don’t provide any systematic framework for discovering the correct prompt template beyond prompt engineering, which limits the paper’s practicality.

**Questions:**

**L280**: Why does the answer space in chess have infinite cardinality? The number of answers is bounded, so it should not be infinite.

**L306**: "would expand the search space nearly to len(CR)/len(S) , as the number of pieces doesn’t provide useful information for determining the next valid move. Could the authors clarify their definition of "useful" information? For example, if only two kings remain, it’s clear that the game is over.

**Table 2**: could you clarify (average) number of output tokens (steps) taken for each technique

---

### Official Review · Reviewer_A1pt · 2024-11-04

**Soundness:** 2
**Presentation:** 1
**Contribution:** 2
**Rating:** 3
**Confidence:** 4

**Summary:**

This paper examines the limitations of Transformer-based Large Language Models (LLMs) in handling complex reasoning tasks, due to their limited computational depth. Chain of Thought (CoT) prompting, while promising, often applies a "one-prompt-for-all" approach across diverse tasks, which can hinder the model’s ability to generate accurate reasoning steps. The authors build on theoretical analyses of CoT, proposing that task-specific supervision is necessary to effectively navigate the prompt space and enhance reasoning accuracy. Through experiments, they demonstrate that supervised CoT—enriched with human guidance—improves performance over unsupervised prompting, advocating for a more tailored approach to maximize LLMs’ reasoning capabilities.

**Strengths:**

The paper effectively addresses the limitations of Transformer-based LLMs in handling complex reasoning tasks, particularly due to computational depth constraints, making a strong case for exploring alternatives like CoT prompting.

By analyzing Transformer limitations and explaining how CoT extends reasoning capabilities, the paper provides a robust theoretical foundation, linking CoT’s potential to overcoming computational constraints.

Introducing task-specific “supervised” CoT as an alternative to the “one-prompt-for-all” method is innovative, aiming to enhance reasoning accuracy and flexibility by tailoring prompts to specific tasks.

The paper’s emphasis on refining prompt design and reasoning strategies to address current gaps in CoT applications highlights its real-world relevance and potential for impactful advancements in LLM reasoning.

**Weaknesses:**

While the theoretical foundation of CoT and supervised prompting is well-explained, there is little mention of extensive empirical results to support the effectiveness of the proposed supervised CoT approach. Without comprehensive experiments, the practical impact and robustness of the approach may be less convincing.

The introduction is densely packed with technical details and may be challenging for readers unfamiliar with computational depth limitations and CoT theory. Simplifying or clarifying certain concepts could improve accessibility for a broader audience.

Although task-specific “supervised” CoT is presented as an improvement over unsupervised CoT, it requires additional human guidance, which may limit scalability and applicability in cases where expert supervision is costly or unavailable.

The paper could be strengthened by comparing supervised CoT with other advanced prompting or finetuning techniques that address reasoning depth or diversity. This would contextualize the performance of supervised CoT against other methods and provide clearer insights into its unique benefits and trade-offs.

More importantly, the presentation and readability of this paper may be far below what ICLR requires. For instance, every figure in this paper is extremely blurry and poorly designed.

**Questions:**

See Weaknesses Above.

---

### Note · Authors · 2024-12-03

I have read and agree with the venue's withdrawal policy on behalf of myself and my co-authors.